# Climate indices for Baltic States from principal component analysis

Liga Bethere[1], Juris Sennikovs[1], Uldis Bethers[1]

[1]Laboratory for Mathematical Modelling of Environmental and Technological Processes, University of Latvia, Riga, LV-1002, Latvia

*Correspondence to*: Liga Bethere (liga.bethere@lu.lv)

**Abstract.** We used principal component analysis (PCA) to derive climate indices that describe the main spatial features of the climate in the Baltic States (Estonia, Latvia and Lithuania). Monthly mean temperature and total precipitation values derived from the ensemble of bias-corrected regional climate models (RCM) were used. Principal components were derived for years 1961-1990. The first three components describe 92% of the variance of the initial data and were chosen as climate

indices in further analysis. Spatial patterns of these indices and their correlation with the initial variables were analyzed and it was detected (based on correlation coefficient between principal components and initial variables) that higher values of each index corresponded to locations with: (1) less distinct seasonality, (2) warmer climate and (3) wetter climate. In addition for the pattern of the first index impact of the Baltic Sea (distance to coast) was apparent, for the second – latitude and elevation, and for the third - elevation. The loadings from the chosen principal components were further used to

calculate values of the climate indices for years 2071-2100. Overall increase was found for all three indices with minimal changes in their spatial pattern.

## 1 Introduction

Spatial representation of the climate e.g. mapping of climatic zones is a useful tool in climate analysis. First, it can be used to better convey information about the climate features of the region, for applications in climate change adaptation and

mitigation. Second, the spatial patterns can give insight both into the possible relationship and the impact of the climate on other fields, e.g., phenological processes and vegetation distribution (Feng et al. 2012). Third, they illustrate geographical features that influence climate, such as hillsides, coastal zones etc. There is a wide variety of approaches for creating spatial representations of climate, but usually they belong to either rule-driven or data-driven methods. Rule-driven methods are used more often, the most popular being the Köppen-Geiger classification (Peel et al. 2007). These methods are based on

some predefined rules, for example, thresholds of meteorological variables or frequency of events. Climate zones derived from classifications of this type usually correspond to vegetation distributions, in the sense that each climate type is dominated by one vegetation zone or eco-region (Belda et al. 2014). However, predefined rules make these methods subjective. Alternatively, the spatial pattern can be derived from data-driven or analytical methods. These include principal component analysis (PCA, Benzi et al. 1997, Estrada et al. 2008) and cluster analysis (Bieniek et al. 2012) or a combination

of both methods (Briggs and Lemin 1992, Fovell and Fovell 1993, Baeriswyl and Rebetez 1997, Malmgren et al. 1999, Fan

et al. 2014, Forsythe et al. 2015). Analytical methods, depending on the chosen variables, can give results that are similar to those of rule-driven methods, but results are more homogenous (Netzel and Stepinski 2016). Analytical methods provide a spatial pattern that must be interpreted before it can be linked with possible applications.

Principal component analysis or empirical orthogonal function analysis has two important applications. First, it can reduce the number of variables that are used to describe regional climate while still retaining most of the variation seen in the initial data. Second, principal components provide new indices that are a linear combination of the chosen variables. The loadings of the chosen principal components are the coefficients that define the newly created indices, which then describe the main features of climate. Variables for PCA can be chosen and indices calculated with a specific purpose in mind, for example, indices for the classification of different types of winters (Hagen and Feistel 2005) or estimation of crop yield based on the climate (Cai et al. 2013). Indices can also be chosen to describe the climate of the region in general (Estrada et al. 2008). However, the problem with the indices that are derived using analytical methods is that their meaning is not known beforehand, so their interpretation may require further analysis.

For many practical applications temperature and precipitation are the two main variables of interest for a certain region. They are usually sufficient for representing vegetation types in corresponding climate zones (Zhang and Yan 2014). Vegetative production, organic matter decomposition, and cycling of nutrients are strongly influenced by temperature and moisture (Briggs and Lemin 1992). Distinct changes of temperature and precipitation are to be expected in future (BACC 2015). Thus, consequently, any climate patterns based on these two variables also will be affected, leaving a significant impact on living organisms. For instance, plant species inhabiting regions subjected to climate change might have too little time to adapt (Mahlstein et al. 2013).

The Baltic States region exhibits significant spatial and temporal climatic variability, with influence of air masses from arctic to subtropical origin (Jaagus and Ahas 2000, Rutgersson et al. 2014). The terrain is mostly flat, with the highest elevations extending slightly above 300 meters. The Baltic Sea and the shape of its coastline have an important role in the climate of the region. PCA has been used to describe precipitation pattern in the Baltic countries with atmospheric and landscape variables (Jaagus et al. 2010).

To study the effects of climate change on climate patterns regional climate model (RCM) data can be used (Castro et al. 2007, Mahlstein and Knutti 2010, Tapiador et al. 2011, Fan et al. 2014). RCM models are continuously improving and correspond rather well to climate observations (Tapiador et al. 2011). Other advantages of using RCM data are that (a) their data are regularly spaced while PCA applied on irregularly spaced data can produce distorted loading patterns (Karl et al. 1982) and (b) RCM data are available also as future projections giving insight into manifestation of climate change. Additionally, the spatial representativeness of the network of observation stations in the Baltic States has been reported to be problematic (Remm and Jaagus 2011).

The aim of this work is to define climate indices which represent the main features of Baltic States climate in a compact form. The study consists of several parts. First, RCM data for temperature and precipitation were bias-corrected. Second, monthly average values for the reference period 1961-1990 were calculated and standardized. Third, PCA analysis was

performed and main principal components were identified. Acquired principal components and their spatial patterns were analyzed. Fourth, loadings of chosen principal components were used to calculate indices for years 2071-2100 and compared to reference data.

## 2 Data and Methods

### 2.1 Climate data and methods

The source of the RCM ensemble data is the ENSEMBLES project (van der Linden and Mitchell 2009). Model data sets for the A1B scenario are given for the time period 1961 – 2100. 22 model runs were considered (shown in Table 1).

We used time series of daily average air temperature at 2m height and daily precipitation. RCM models are known to be prone to systematic biases (Teutschbein & Seibert 2012). A bias correction method (Sennikovs and Bethers 2009) that uses quantile mapping was chosen and the cumulative distribution function was calculated for each day of the year using 11-day running average – the data for five days before and five days after the day of interest. The ensemble median was then used for PCA. The control period for bias correction was 1961-1990. Bias corrected data was then interpolated to a regular grid because it has been shown that PCA applied to irregularly spaced data can produce distorted loading patterns (Karl et al. 1982). The bias correction method and model resolution is described in detail in Sennikovs and Bethers, 2009.

Two time periods were chosen – 1961-1990 (as reference climate) and 2071-2100 (as future climate projections). For each time period monthly average temperature and precipitation were calculated for each grid point. In total 24 climatic variables were used for each time period - 12 monthly precipitation and 12 monthly average temperatures. This is an "R-mode" analysis according to Cattell (1952). The spatial distribution of these variables for the reference period is shown in Fig. 1 and Fig. 2. Figure 1 shows a north-south gradient of monthly precipitation during April-June and east-west gradient of monthly precipitation during October-January. Figure 2 shows an east-west gradient of monthly temperatures during October-February and north-south gradient of monthly temperatures during April-June. This implies that some of the variables can be combined in seasons (as it is done by Malmgren et al. (1999) and Forsythe et al. (2015)) and that for some months temperature and precipitation is correlated. A better understanding of variables with similar patterns can be gained by examining the correlation matrix in Fig. 3. The matrix areas that represent strongly correlated variables are marked in this figure and they show the following relationships:

1 - Very strong correlation (above 0.8) between precipitations in winter months – locations with more precipitation in, e.g., December also have more precipitation in January (compared to the rest of the territory).

2 - Strong correlation (above 0.5) between precipitation and temperature in spring months. Thus, locations with colder springs also are dryer, whilst locations with warmer springs also have more spring precipitation.

3 - Strong negative correlation (below -0.5) between precipitation in autumn and late spring/early summer temperature – locations with more precipitation in autumn also have colder spring.

4- Very strong correlation (above 0.8) between temperatures of autumn and winter months – locations with warmer autumn also have warmer winter.

Figure 3 shows that 24 monthly variables contain redundant information and through PCA we can summarize information and create new variables.

## 2.2 PCA method

The aim of PCA is to create a new set of uncorrelated variables that are a linear combination of the initial variables and explain as much as possible of the initial variation. An extensive description of PCA can be found in Jolliffe (2002), and its applications to climate are described in Preisendorfer (1988).

Although PCA is a widely used methodology, the terminology in literature can vary (Wilks, 2011). We will briefly describe
the terminology used in this article.

Suppose that $X$ is an $n \times p$ data matrix, where $n$ is the number of objects and $p$ is the number of variables. The means of the $p$ variables have been subtracted. In our case we have $p = 24$ climatic variables in $n = 7143$ grid points. A typical PCA is applied to $p \times p$ covariance (or correlation) matrix calculated by Eq. (1). Then by solving Eq. (2) we can find eigenvectors $e_i, i = 1, \dots, 24$ and corresponding eigenvalues $\lambda_i, i = 1, \dots, 24$. As a result we have obtained non-correlated linear
combinations of initial climatic variables calculated by Eq. (3).

$$S = (n - 1)^{-1} X^T X. \tag{1}$$

$$Se = \lambda e \tag{2}$$

$$Y_i = X_i e_i, \quad i = 1, \dots, 24. \tag{3}$$

Values $\lambda_i$ represent explained variance of each "principal component" $Y_i$. Linear weights $e_i$ that define each principal
component will be called "loadings". "Indices" describe $Y_i$ that are calculated using loadings from reference period (but not necessarily reference period data). For the reference period principal components coincide with indices, but indices can be also calculated using future period data and reference period loadings.

An important choice must be made when applying PCA: whether to use correlation matrix or covariance matrix in the calculation of loadings. If the covariance matrix is used then a second choice must be made – if and what standardization to
25 use. Scaling process has a significant impact on the PCA process. When performing data standardization following issues should be taken into account:

1 – Variables should be of similar scale, otherwise variables with considerably larger variance will dominate the principal components. Different scales are usually a consequence of different units of measurement. In our case the variance for precipitation measured in millimeters is considerably larger than that of temperature that is measured in degrees Celsius.
30 2 – In case of variables that are measured in the same units variances contain useful information and can improve the interpretation of PCA (Overland and Preisendorfer 1982). Therefore, for variables that are measured in the same units (for example, average temperature of different months) we wish to keep the ratio between variances of different months. This

means that the correlation matrix, where each variable is divided by its square root of variance, should not be used, as it would bring the variances of all 24 variables to 1.

3 – As we are planning to use the acquired loadings as coefficients for the calculation of climate indices for the future time period and compare them with the reference climate it is necessary that the same standardization process is used for the data of the future time period.

4 – It is important to note that subtraction of mean (or similar constant) for each variable does not impact the result of PCA as it does not impact the covariance between variables. However if the initial values have zero mean (mean is subtracted from each variable) then the resulting principal components have similar scale and spatial patterns are more convenient to review.

Taking into account the issues described above we propose to use standardization as defined by Eq. (4), where the spatial mean is subtracted for each variable as usual, but the average variance of all temperature or precipitation variables is used for scaling:

$$\frac{T_k - \bar{T}_k}{\sqrt{\bar{V}(T)}}, \qquad \frac{P_k - \bar{P}_k}{\sqrt{\bar{V}(P)}}, \qquad k = 1, \dots, 12, , \tag{4}$$

where $\bar{V}(T), \bar{V}(P)$ – average variance of 12 temperature and precipitation variables for reference period.

The variances before and after such standardization for reference period are shown in Table 2. The ratio of variances for different months is retained. For data representing the future time period the standardization is performed by using the mean values and average variances from the reference period. Results of data standardization for the future time period are shown in Table 3. It can be seen that in the future the variance of precipitation data will increase and the variance of temperature data will decrease. However, the distribution of variances over the year is similar.

Another detail that must be considered when using PCA is the choice of method for determining the number of principal components that describe data variation sufficiently well and can be used in further analysis. There are multiple methods to choose from (Preisendorfer 1988), however in our case one of the most common methods – scree-plot – gives excellent and clear results. A scree plot is a graph of explained variances of acquired principal components and the number of principal components is decided based on the break point in such a graph. Components to the left of the break point are retained.

## 3 Results

### 3.1 Principal components for the control period (1961-1990)

Explained variance and loadings of the first 3 principal components are shown in Table 4. The scree-plot of all principal components is shown in Fig. 4. The first two components already describe 78% of the variance of initial variables, while the first three components describe 92% of the variance. According to Jolliffe (2002) the cutoff point should be between 70%

and 90% of the explained variance. However, the scree-plot clearly shows that first 3 principal components can be retained, so we chose to further analyze the first 3 components.

Figure 5 shows the spatial pattern of the first three principal components for the reference climate. They should be analyzed together with the correlation coefficients between the new variables and initial variables shown in Table 5, where the bright

red or blue colors mark high positive or negative correlation. One can see that variables that were initially highly correlated (positively or negatively, Fig. 3) show similar (or in case of negative correlation – opposite) values in Table 5.

Correlation coefficient values (Table 5) show that the first principal component (PC1) has a high positive correlation with the autumn-winter temperature and precipitation and a high negative correlation with temperature and precipitation in late spring and early summer months. This means that higher values of PC1 correspond to places with warmer winters with more

precipitation (snow or rain) and colder summers with less precipitation. However it is also important to note, that the total sum of loadings is above 1, which implies that a constant increase in all variables would also result in higher values of PC1. From the spatial distribution (Fig. 5) we can see that PC1 has an east – west gradient implying less distinction between seasons at the seaside. It can be concluded that PC1 reflects the continentality of climate, it represents the influence of the Baltic Sea.

The second principal component (PC2) is positively correlated with all monthly temperatures and negatively correlated with precipitation in autumn. This means that high PC2 values correspond to regions that are generally warmer than others and have low precipitation in autumn. For PC2 a north – south gradient is evident with the warmer climate in south. This means that PC2 represents the influence of latitude. This pattern is also slightly influenced by geographical features (elevation) and the shape of the coast.

PC3 is mainly positively correlated with precipitation for most of the year (December – August) and spring temperature (April – May). This means that high PC3 values correspond to places with overall high precipitation, or in other words – an overall wetter year. PC3 mainly reflects the terrain, i.e. the distribution of elevation.

When the spatial patterns of PC2 and PC3 are analyzed the effect of orography can be seen, especially, the location of highlands is clearly visible, while for PC1 the terrain seems to have little impact.

**3.2 Climate indices for future climate (2071-2100)**

Loadings (linear weights) acquired through PCA from reference data (Table 4) can be used as coefficients that define new climate indices. We can use these coefficients to calculate climate from different data (other time period or other geographical locations). It is also important to note that statistics (mean values and variances) from reference data used in data standardization should be applied to other data as well for comparison to be possible. In our case we calculated such

climate indices for future climate (corresponding to period 2071-2100) and analyzed the change in climate patterns. Standardization of variables is shown by Eq. (5) and calculation of climate indices by Eq. (6).

$$\frac{T_k - \bar{T}_k}{\sqrt{\bar{V}(T)}}, \qquad \frac{P_k - \bar{P}_k}{\sqrt{\bar{V}(P)}}, \qquad k = 1,\dots,12,, \tag{5}$$

where $T_k, P_k$ – temperature and precipitation values for future period, $\bar{T}_k, \bar{P}_k$ - mean temperature and precipitation values for the reference period, $\bar{V}(T), \bar{V}(P)$ – average variance of 12 temperature and precipitation variables for the reference period.

$$Y_i = X_i c_i, \quad i = 1, \dots, 24, \tag{6}$$

where $X_i$ – temperature and precipitation data for future period, $c_i$ – coefficients (loadings) from the reference period, $Y_i$ – climate indices for future period.

It is important to note that $Y_i$ should not be called "principal components" even though they hold similar meaning as principal components from reference data. $Y_i$ are not derived using PCA directly and they do not use eigenvectors from future data.

In Fig. 6 the correlation coefficients between indices and initial variables are shown and it can be seen that they are similar to those for past climate. Therefore, they have the same interpretation and it is possible to analyze the change in spatial patterns between the past and future climate. The spatial distributions of future indices are shown in Fig. 7. Statistical descriptors, e.g., minimal, maximal and mean value of past and future indices are summarized in Table 6. In addition, as we have used the same standardization (subtraction of reference period mean) and climate indices calculation process (loadings from reference period) we can derive conclusions about increase or decrease of these climate indices. However it is important to note that no conclusions can be derived about the value by which the increase/decrease has happened.

All indices have higher values in future climate. This can be interpreted as an overall warmer climate (increase of PC2) and wetter climate (increase of PC3). Interpretation of PC1 is more complicated as coefficients (Table 4) for some variables are positive and for some are negative. Increase in PC1 would be observed in case of constant increase in all variables. However, an increase would also be observed in case of temperature and precipitation decrease in spring/summer. An average increase of "standardized" (by Eq. (5)) mean values is 1.4 units for temperature and 4.5 units for precipitation. Such a constant increase with coefficients in Table 4 would result in 6.5 unit increase for PC1. As we can see from index statistics in Table 7 – increase of 8.4 units is observed for PC1, so we suspect that the additional increase can be attributed to changes in seasonality.

For PC1 it is shown that the values corresponding to coastal regions in the reference climate will "move" to the eastern part of Baltic States in the future projections. The expected changes of PC2 are the largest – the maximum values of PC2 for the reference climate (in southern Lithuania) are lower than the minimum values for the future climate (in central Estonia). Statistics in Table 6 show that the reference range of this index does not overlap with the range of future values. The climate corresponding to reference values of PC3 in western Lithuania (Zhemaichiai highland) will in future be observable in plateaus in central and north-eastern part of Baltic States.

## 4 Discussion

The methodology used in this study has been able to reduce 24 climate variables to 3 new indices that more efficiently and compactly represent the main features of the climate in the Baltic countries. The methodology can also be applied to the

future climate data and therefore the impacts of climate change can be analyzed. Additional analysis is needed for the interpretation of the acquired indices.

Some insight into the possible interpretation of the acquired climate indices can be gained from the literature. The spatial distribution of PC1 is similar to the spatial patterns of mean start date of winter (see results for Estonia in Jaagus and Ahas
(2000)) with higher PC1 values corresponding to later winters.

As PC2 is mainly linked to temperature, the patterns exhibited by PC2 can be expected to be similar to the spatial distribution of phenological events for which the temperature is the main driving factor. For example, the spatial pattern of PC2 shows similarities to spring and summer start dates in the Baltic Sea region and to more specific phenological events, such as apple tree blossoming and beginning of the vegetation of rye (Jaagus and Ahas 2000) or strawberry blooming and
harvest (Bethere et al. 2016). In general, higher values of PC2 correspond to places with earlier phenological processes.

High values of winter precipitation and high temperatures in spring can be interpreted in the context of spring floods – however additional analysis is needed to account for the snow cover. The spatial distribution of PC3 is similar to the map of average annual precipitation in the study region (Jaagus et al. 2010). Interestingly, the precipitation in autumn months (September – October) has a small contribution to PC3 (Table 5).
Conclusions based on spatial pattern and correlation coefficient analysis are summarized in Table 7.

The methodology could be further improved to better link acquired indices with phenological processes or seasons by either rotating acquired principal components (Jolliffe 2002) or performing correlation or regression analysis with other variables, such as crop yield (Cai et al. 2013). This approach would be especially useful in case of PC1, where currently analysis is complicated due to both changes in seasonality and the constant increase affecting PC1 values. Another approach that could
be used to describe the spatial variability of climate in the Baltic States is clustering based on the chosen principal component values (Fovell and Fovell 1993, Forsythe et al. 2015).

If variables other than temperature or precipitation are used for the principal components analysis, in some cases the standardization procedure should be modified. However, it should be taken into account that when more than one data set is used, e.g., when past and future climate is compared, the same values used for standardization should be applied to all of
them.

**5 Conclusions**

Most of the spatial variability of monthly average temperature and precipitation over the Baltic countries can be represented by three principal components both for past and future climate. These components can be considered as climate indices, where higher values of each index correspond to locations with (1) climate with less distinct seasons, (2) warmer climate, (3)
climate with more precipitation. Each component has a distinct spatial pattern. The index related to seasonality exhibits a clear east-west (or inland) gradient with less distinct seasonality at seaside (West). The second index (warmer climate) shows a north-south gradient with a warmer climate in south. This index also reflects orography with colder climate in hilly

regions. The third index reflects the overall precipitation. Its spatial distribution is mainly dominated by elevation, with maxima at the highlands and less precipitation in plains and at the seaside. Specific standardization of data allows calculation of such indices also for the future climate. Change in the climate indices in the future implies less distinct seasons, warmer and wetter climate.

5    Although there is significant change in the magnitude of the indices between future and reference period, the change in spatial distribution is relatively small. For the first and third component regions can be identified where future climate will be similar to the climate currently in other regions.

**Acknowledgments**

The research was supported by the Latvian State Research Programme "The value and dynamic of Latvia's ecosystems under changing climate" (EVIDEnT).

The ENSEMBLES data used in this work was funded by the EU FP6 Integrated Project ENSEMBLES (Contract number 505539) whose support is gratefully acknowledged.

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

# Figures and Tables

**Figures**

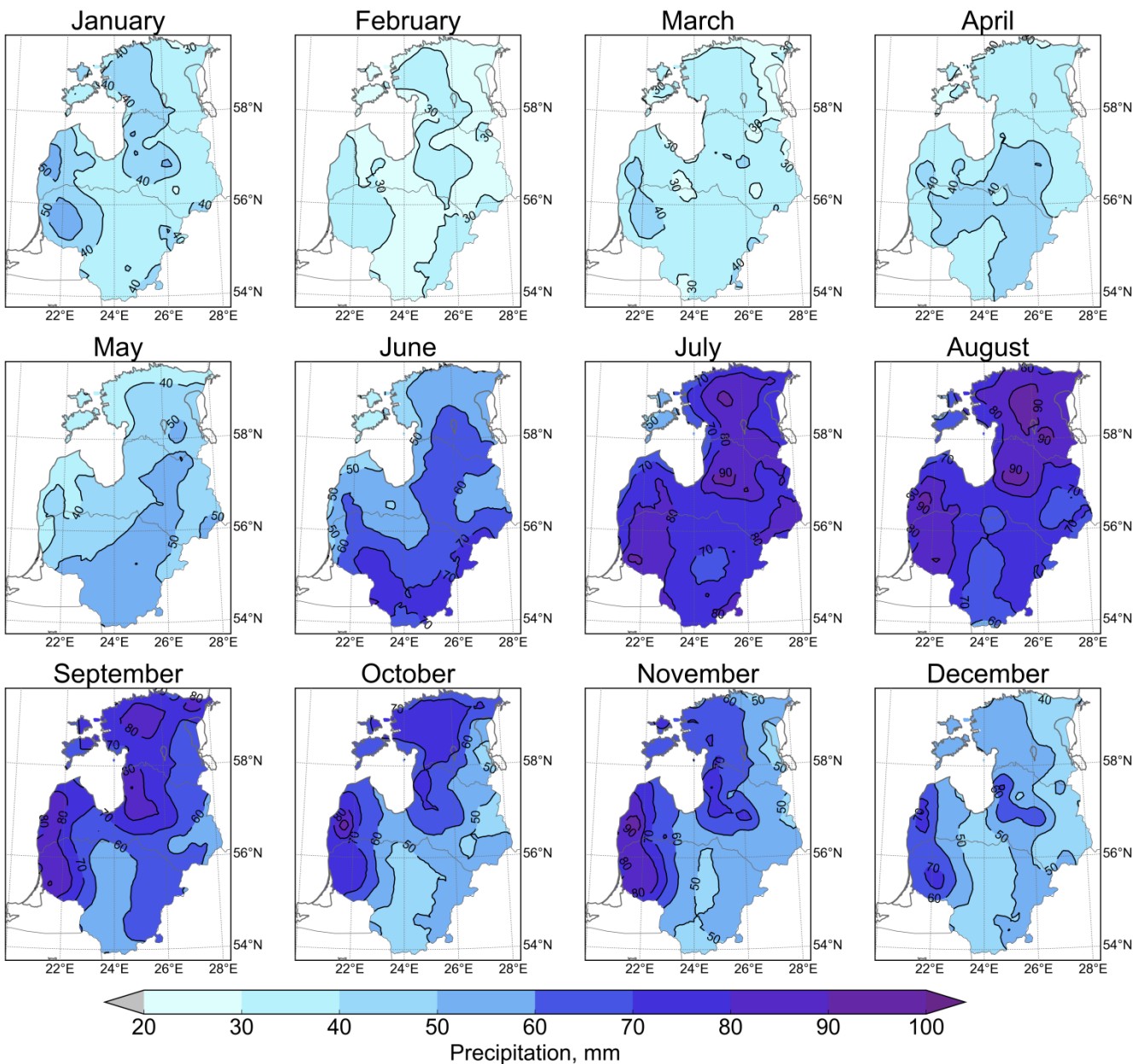

Figure 1: Monthly precipitation 1961-1990, bias-corrected median of RCM ensemble.

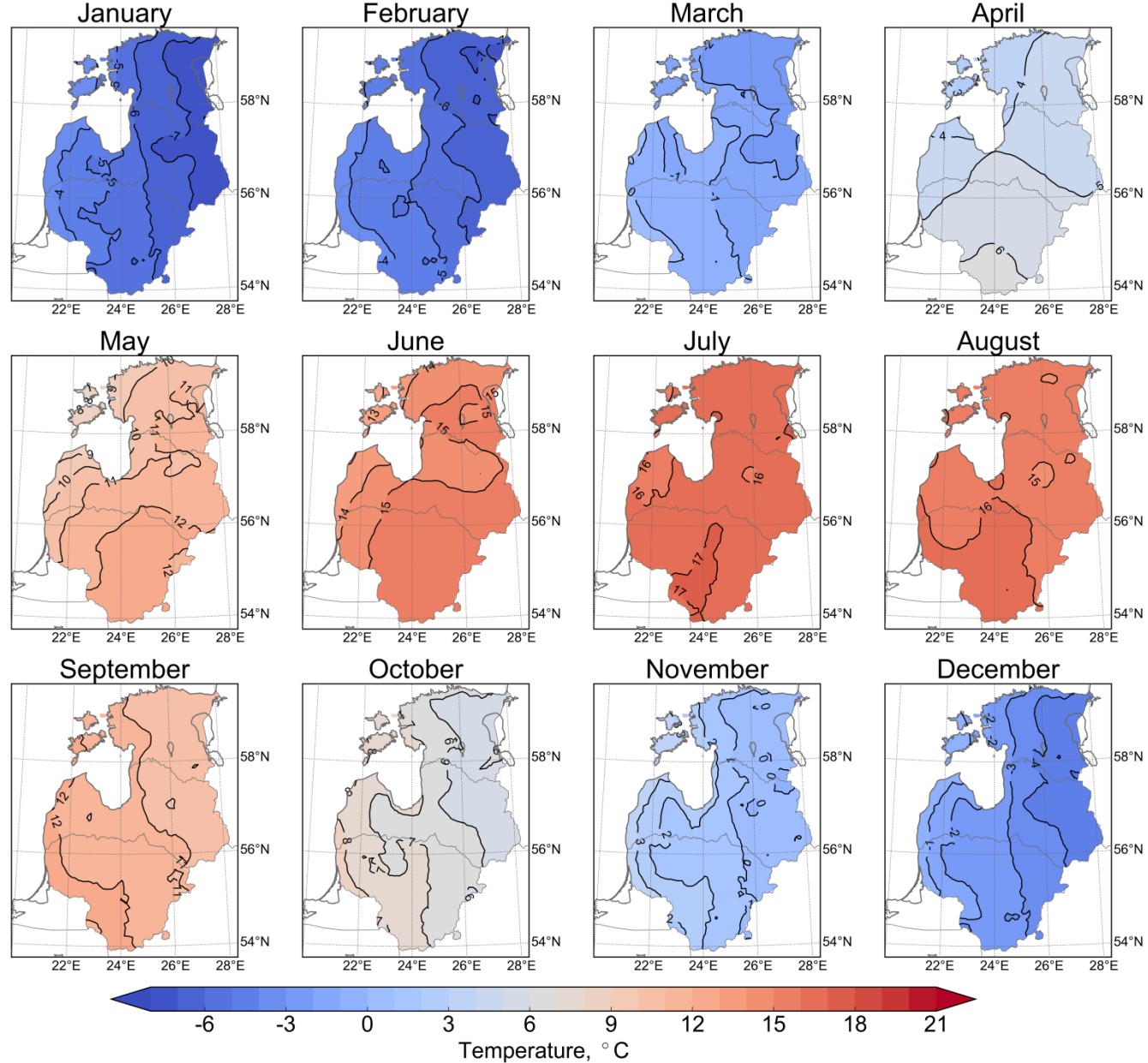

**Figure 2: Monthly average temperature 1961-1990, bias-corrected median of RCM ensemble.**

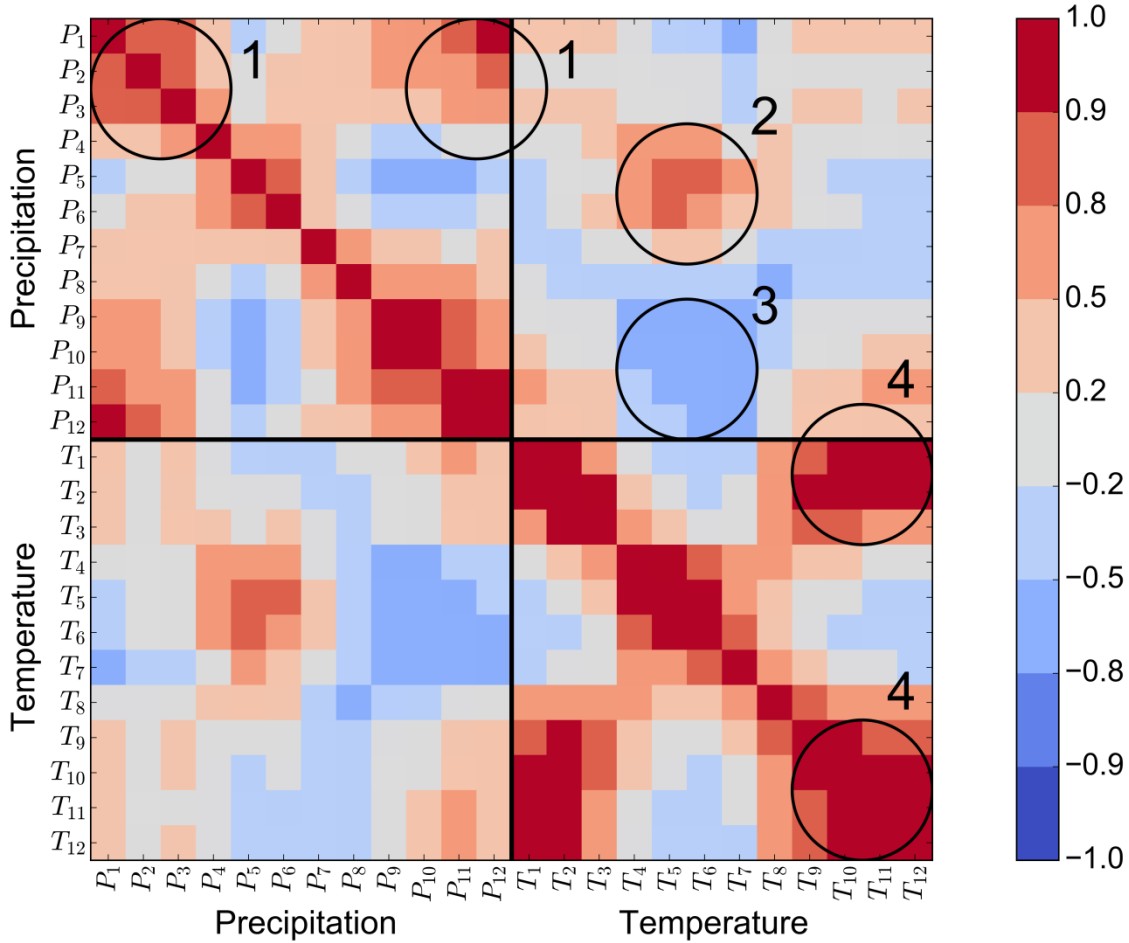

**Figure 3: Temperature-precipitation correlation matrix, bias-corrected data.** Marked and numbered features show especially high absolute correlation: 1- strong correlation between precipitation in winter months; 2- strong correlation between precipitation and temperature in spring months; 3- strong negative correlation between precipitation in autumn and spring temperature; 4- strong correlation between temperatures of autumn and winter months.

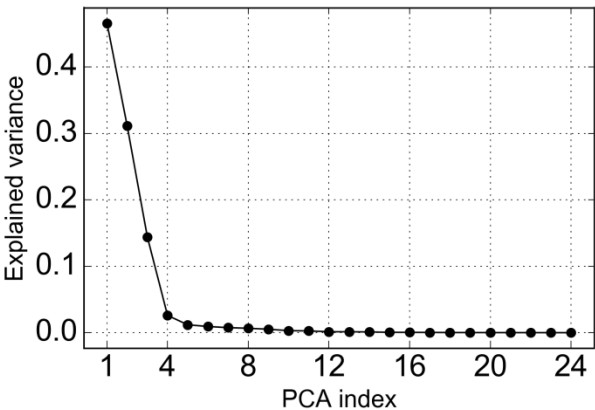

**Figure 4: Scree plot (explained variance of each principal component), calculated for reference (1961-1990) climate.**

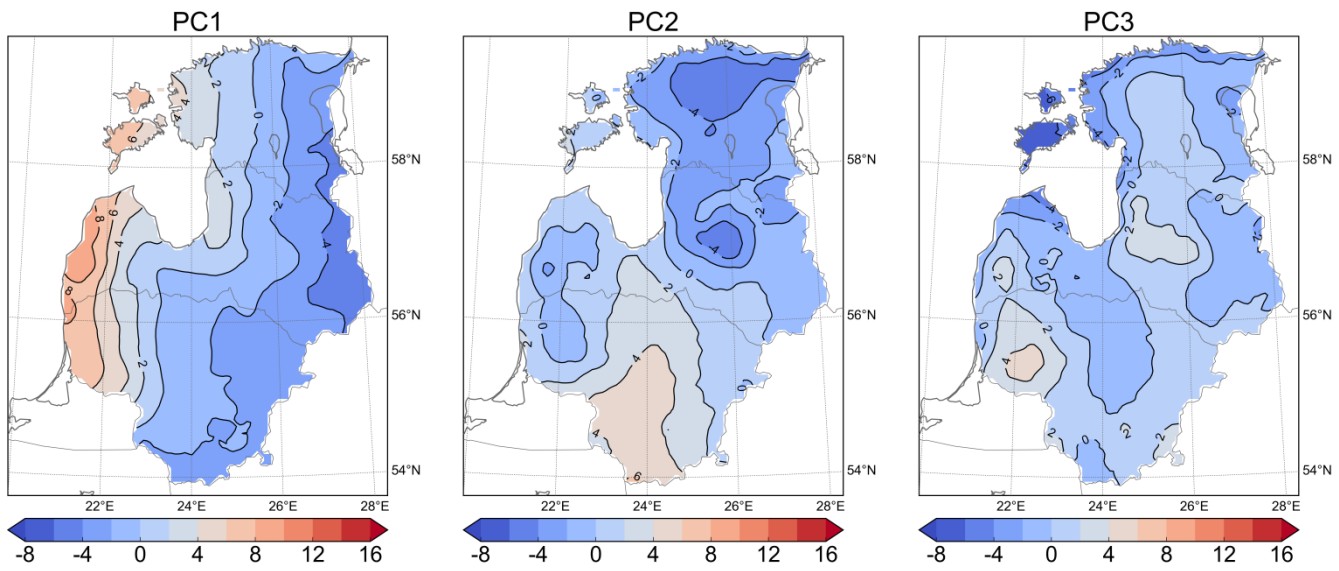

**Figure 5: Spatial pattern of first three principal components based on monthly temperature and precipitation data for years 1961-1990.**

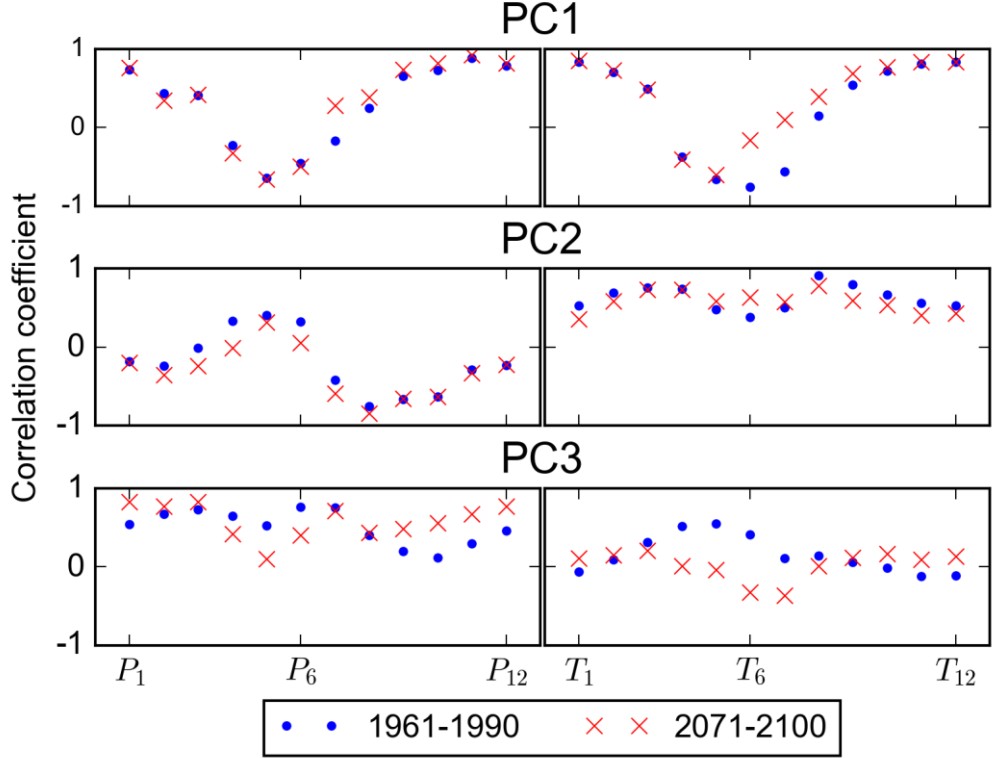

**Figure 6: Correlation coefficients between indices (principal components) and initial variables for reference and future climate.**

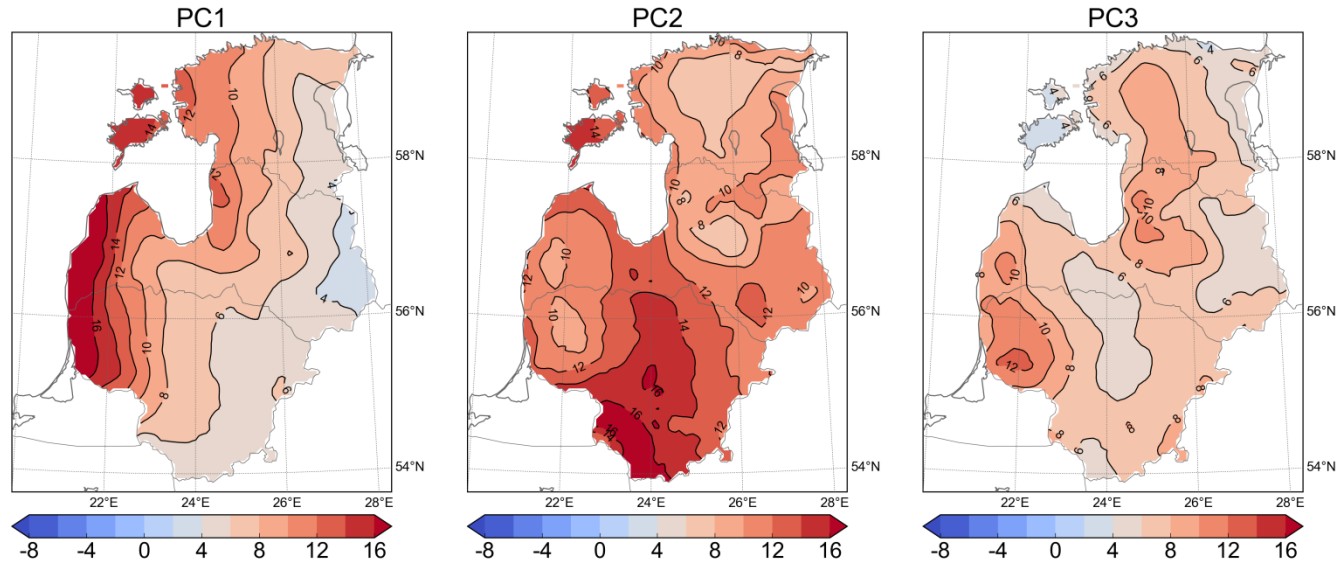

5 **Figure 7: Climate indices (based on principal components from 1961-1990) for years 2071-2100.**

**Tables**

**Table 1: List of the Regional Climate Model (RCM) ensemble members used (ENSEMBLES), showing the originating institution, the name of RCM, the driving General Circulation Model (GCM). For explanation of abbreviations see van der Linden & Mitchell (2009).**

| Institution | GCM | RCM |
| --- | --- | --- |
| C4I | HadCM3Q16 | RCA3 |
| CNRM | ARPEGE | Aladin |
| CNRM | ARPEGE_RM 5.1 | Aladin |
| DMI | ARPEGE | HIRHAM |
| DMI | ECHAM5-r3 | DMI-HIRHAM5 |
| ETHZ | HadCM3Q0 | CLM |
| GKSS | IPSL | CLM |
| HC | HadCM3Q0 | HadRM3Q0 |
| HC | HadCM3Q16 | HadRM3Q16 (high sensitivity) |
| HC | HadCM3Q3 | HadRM3Q3 (low sens.) |
| ICTP | ECHAM5-r3 | RegCM |
| KNMI | ECHAM5-r3 | RACMO |
| KNMI | ECHAM5-r3 | RACMO |
| KNMI | MIROC | RACMO |
| METNO | BCM | HIRHAM |
| METNO | HadCM3Q0 | HIRHAM |
| MPI | ECHAM5-r3 | REMO |
| SMHI | BCM | RCA |
| SMHI | ECHAM5-r3 | RCA |
| SMHI | HadCM3Q3 | RCA |
| UCLM | HadCM3Q0 | PROMES |
| VMGO | HadCM3Q0 | RRCM |

**Table 2: Variances of climate variables before and after standardization for years 1961-1990.**

| 1961-1990 | | | | | | | | | | | | |
| --- | --- | --- | --- | --- | --- | --- | --- | --- | --- | --- | --- | --- |
| Before standardization | | | | | | | | | | | | |
| $P_1$ | $P_2$ | $P_3$ | $P_4$ | $P_5$ | $P_6$ | $P_7$ | $P_8$ | $P_9$ | $P_{10}$ | $P_{11}$ | $P_{12}$ | Mean |
| 28.85 | 7.45 | 13.03 | 13.66 | 31.93 | 63.40 | 47.20 | 65.65 | 86.22 | 110.43 | 114.47 | 50.60 | 52.74 |
| $T_1$ | $T_2$ | $T_3$ | $T_4$ | $T_5$ | $T_6$ | $T_7$ | $T_8$ | $T_9$ | $T_{10}$ | $T_{11}$ | $T_{12}$ | Mean |
| 1.36 | 0.95 | 0.60 | 0.62 | 0.93 | 0.41 | 0.09 | 0.19 | 0.39 | 0.54 | 0.83 | 1.27 | 0.68 |
| After standardization | | | | | | | | | | | | |
| $P_1$ | $P_2$ | $P_3$ | $P_4$ | $P_5$ | $P_6$ | $P_7$ | $P_8$ | $P_9$ | $P_{10}$ | $P_{11}$ | $P_{12}$ | Mean |
| 0.55 | 0.14 | 0.25 | 0.26 | 0.61 | 1.20 | 0.89 | 1.24 | 1.63 | 2.09 | 2.17 | 0.96 | 1.00 |
| $T_1$ | $T_2$ | $T_3$ | $T_4$ | $T_5$ | $T_6$ | $T_7$ | $T_8$ | $T_9$ | $T_{10}$ | $T_{11}$ | $T_{12}$ | Mean |
| 2.00 | 1.40 | 0.88 | 0.91 | 1.37 | 0.60 | 0.14 | 0.27 | 0.57 | 0.80 | 1.22 | 1.86 | 1.00 |

**Table 3: Variances of climate variables before and after standardization for years 2071-2100.**

| 2071-2100 | | | | | | | | | | | | |
| --- | --- | --- | --- | --- | --- | --- | --- | --- | --- | --- | --- | --- |
| before standardization | | | | | | | | | | | | |
| $P_1$ | $P_2$ | $P_3$ | $P_4$ | $P_5$ | $P_6$ | $P_7$ | $P_8$ | $P_9$ | $P_{10}$ | $P_{11}$ | $P_{12}$ | Mean |
| 52.78 | 12.33 | 22.68 | 27.02 | 33.84 | 52.5 | 42.87 | 72.7 | 126.1 | 154.3 | 204.3 | 85.6 | 73.92 |
| $T_1$ | $T_2$ | $T_3$ | $T_4$ | $T_5$ | $T_6$ | $T_7$ | $T_8$ | $T_9$ | $T_{10}$ | $T_{11}$ | $T_{12}$ | Mean |
| 1.08 | 0.92 | 0.37 | 0.25 | 0.26 | 0.12 | 0.11 | 0.2 | 0.45 | 0.51 | 0.84 | 1.08 | 0.52 |
| after standardization | | | | | | | | | | | | |
| $P_1$ | $P_2$ | $P_3$ | $P_4$ | $P_5$ | $P_6$ | $P_7$ | $P_8$ | $P_9$ | $P_{10}$ | $P_{11}$ | $P_{12}$ | Mean |
| 1.00 | 0.23 | 0.43 | 0.51 | 0.64 | 1.00 | 0.81 | 1.38 | 2.39 | 2.93 | 3.87 | 1.62 | 1.40 |
| $T_1$ | $T_2$ | $T_3$ | $T_4$ | $T_5$ | $T_6$ | $T_7$ | $T_8$ | $T_9$ | $T_{10}$ | $T_{11}$ | $T_{12}$ | Mean |
| 1.59 | 1.35 | 0.55 | 0.36 | 0.38 | 0.18 | 0.16 | 0.3 | 0.67 | 0.74 | 1.23 | 1.58 | 0.76 |

**Table 4: Explained variance and loadings of first 3 principal components, calculated from temperature and precipitation data for years 1961-1990.**

|  | PC1 | PC2 | PC3 | sum |
|---|---|---|---|---|
| Explained variance | 0.47 | 0.31 | 0.14 | 0.92 |
| Loadings |  |  |  |  |
| $P_1$ | 0.16 | -0.05 | 0.22 |  |
| $P_2$ | 0.05 | -0.03 | 0.14 |  |
| $P_3$ | 0.06 | 0.00 | 0.20 |  |
| $P_4$ | -0.03 | 0.06 | 0.18 |  |
| $P_5$ | -0.15 | 0.12 | 0.22 |  |
| $P_6$ | -0.15 | 0.13 | 0.45 |  |
| $P_7$ | -0.05 | -0.15 | 0.38 |  |
| $P_8$ | 0.08 | -0.31 | 0.24 |  |
| $P_9$ | 0.25 | -0.31 | 0.13 |  |
| $P_{10}$ | 0.32 | -0.33 | 0.09 |  |
| $P_{11}$ | 0.39 | -0.16 | 0.24 |  |
| $P_{12}$ | 0.23 | -0.08 | 0.24 |  |
| $T_1$ | 0.35 | 0.27 | -0.04 |  |
| $T_2$ | 0.25 | 0.30 | 0.06 |  |
| $T_3$ | 0.14 | 0.26 | 0.16 |  |
| $T_4$ | -0.11 | 0.26 | 0.27 |  |
| $T_5$ | -0.23 | 0.21 | 0.35 |  |
| $T_6$ | -0.18 | 0.11 | 0.17 |  |
| $T_7$ | -0.06 | 0.07 | 0.02 |  |
| $T_8$ | 0.02 | 0.17 | 0.04 |  |
| $T_9$ | 0.12 | 0.22 | 0.02 |  |
| $T_{10}$ | 0.19 | 0.22 | -0.01 |  |
| $T_{11}$ | 0.27 | 0.23 | -0.07 |  |
| $T_{12}$ | 0.34 | 0.27 | -0.08 |  |

**Table 5: Correlation coefficients between principal components and standardized initial data for years 1961-1990. High positive correlation corresponds to darker red color and high negative correlation corresponds to darker blue color.**

|  | PC1 | PC2 | PC3 |
|---|---|---|---|
| $P_1$ | 0.73 | -0.18 | 0.54 |
| $P_2$ | 0.44 | -0.24 | 0.68 |
| $P_3$ | 0.41 | -0.01 | 0.73 |
| $P_4$ | -0.22 | 0.33 | 0.65 |
| $P_5$ | -0.65 | 0.4 | 0.53 |
| $P_6$ | -0.45 | 0.33 | 0.76 |
| $P_7$ | -0.17 | -0.42 | 0.75 |
| $P_8$ | 0.25 | -0.75 | 0.41 |
| $P_9$ | 0.66 | -0.67 | 0.2 |
| $P_{10}$ | 0.73 | -0.63 | 0.12 |
| $P_{11}$ | 0.89 | -0.29 | 0.3 |
| $P_{12}$ | 0.78 | -0.23 | 0.46 |
| $T_1$ | 0.83 | 0.53 | -0.06 |
| $T_2$ | 0.7 | 0.69 | 0.1 |
| $T_3$ | 0.49 | 0.76 | 0.32 |
| $T_4$ | -0.38 | 0.74 | 0.52 |
| $T_5$ | -0.66 | 0.48 | 0.55 |
| $T_6$ | -0.76 | 0.38 | 0.41 |
| $T_7$ | -0.57 | 0.5 | 0.11 |
| $T_8$ | 0.15 | 0.91 | 0.14 |
| $T_9$ | 0.54 | 0.8 | 0.06 |
| $T_{10}$ | 0.72 | 0.67 | -0.01 |
| $T_{11}$ | 0.81 | 0.56 | -0.12 |
| $T_{12}$ | 0.83 | 0.53 | -0.11 |

**Table 6: Statistics of climate indices (based on PCA) for past and future data.**

|     |      | 1961-1990 | 2071-2100 |
|-----|------|-----------|-----------|
| PC1 | mean | 0.00      | 8.38      |
|     | min  | -4.84     | 3.17      |
|     | max  | 8.95      | 18.24     |
| PC2 | mean | 0.00      | 11.38     |
|     | min  | -5.62     | 6.24      |
|     | max  | 6.14      | 17.05     |
| PC3 | mean | 0.00      | 7.13      |
|     | min  | -8.43     | 1.54      |
|     | max  | 4.84      | 12.28     |

**Table 7: Description and interpretation of climate indices base on PCA.**

| Name | High values correspond to locations with | Possible interpretation of high values |
|------|------------------------------------------|----------------------------------------|
| PC1  | Warm winter with high precipitation, cold summer with low precipitation | Less distinct seasonality |
| PC2  | High overall temperature, low precipitation in autumn | Warmer climate |
| PC3  | High annual precipitation, warmer springs | More humid climate |

