# Peer review of "Climate indices for Baltic States from principal component analysis"

_Earth System Dynamics, 2017_

## Referee Comment (RC1) · Anonymous Referee #1 · 17 Apr 2017

**Referee comments to the manuscript**

**Climate indices for Baltic States from principal component analysis**

**by Liga Bethere, Juris Sennikovs and Uldis Bethers**

The authors have presented an interesting article concerning climate indices for the Baltic region. Principal component analysis (PCA) was applied for monthly mean air temperature and monthly precipitation, which were derived from the ensemble of bias-corrected regional climate models output for the territory of Estonia, Latvia and Lithuania in 1951-1990. Three main climatic indices were defined and described. The indices were calculated for the period 2071-2100 using the model outputs.

I suggest that the manuscript is worth of publishing but it needs major revision. There are many aspects, which are not sufficiently explained. The interpretation of the results could be improved. My concrete remarks and suggestions are the following.

1. The abstract is too general. I would like to see more concrete results of the study in the abstract.

2. Page 1 line 29. What does mean here the term "homogeneous"? How can the results be more homogeneous? In which sense?

3. Page 2 lines 3-5. This sentence is a bit unclear and confusing to me. It is written the loadings of components are the coefficients that define indices. I have an imagination that loadings of principal components show correlation between time series of the components and observed variables. Can you explain this?

4. Page 2 line 24. Should it be the reference to de Castro et al. 2007?

5. The introduction is lacking of the description of similar studies. PCA is widely used in climatology and also for determining of various climate indices. I suggest a literature overview where is shown how the current study is fitting into other similar studies. The novelty of this study should be clearly indicated.

6. The description of the use of model data could be more precise. Does the ensemble data mean that the averages of 22 model runs were calculated? What was the spatial resolution of the ensemble data? I suppose that the resolution was different for every single model run.

7. It was not clear why the model data were used instead of station data. The density of meteorological stations is rather high in the study area. Therefore, the results of the PCA of station data would be compared with the results of the RCM-based data.

8. Page 3 line 9. It is not indicated from which source the observation data (Fig.1) were obtained.

9. Page 3 line 18. Usually, it is written "…as it is done by Malmgren et al. (1999) and Forsythe et al. (2015)".

10. Page 3 lines 22-23. A very strong correlation in winter precipitation is detected. Is it so that correlations are calculated using the data from the same year? In that case there is not any time lag. I don't believe that there is a correlation above 0.8 between January and December of the same year. I don't believe the statement that winters are either dry or humid. There should be something wrong. I did some calculations with station data of monthly precipitation and did not find any significant correlations. Correlations presented in Fig. 4 are inadequate. Such high correlations in monthly precipitation are not possible at all. All other correlation coefficients seem also suspicious. The reason for presenting the correlation Matrix is not clear.

11. Page 4 line 18. I suggest the word "them" instead of "then".

12. Standardisation of climatic data is a trivial procedure. It is not clear why the variances on tables 2 and 3 are presented in the study. Were they spatial mean variances?

13. I think that more information about the PCA procedure is needed. Was it rotated or non-rotated PCA? There are different modes of PCA: T-mode, S-mode etc. How the matrix was performed? Which were variables and which were cases? What were loadings and what were scores? This information is needed for the interpretation of results.

14. Why the loadings of the three first components are presented in Table 4. What do they show?

15. The spatial patterns of three first components are very interesting and informative. I think that they could be wider and better interpreted. It is clear that PC1 represents the influence of the Baltic Sea. It is the main factor causing spatial climatic differences in the Baltic countries. It is directly related to higher temperature and precipitation in autumn and winter, and lower temperature and precipitation in spring and early summer in the coastal regions. In the hinterland far from the sea the spatial coefficients (scores ?) are negative. In conclusion, PC1 reflects continentality of climate.

16. PC2 reflects the second main factor in formation of climate – i.e. latitude. The pattern shows positive scores in Lithuania and negative scores in Estonia. The southern region is characterised by higher temperature, especially in spring and autumn, comparatively higher precipitation from April to June and lower precipitation during the rest of a year.

17. The spatial pattern of the PC3 is very similar to the mean annual distribution of precipitation in the study region (Jaagus et al. 2010). Two regions with higher precipitation are described by areas of negative coefficients - one in western Lithuania and Latvia, and another in the western part of continental Estonia and central Vidzeme upland in Latvia. Positive areas correspond to coastal regions with lower precipitation in Estonia and Latvia. But I cannot

understand why spatial coefficients (loadings) on Fig. 6 are negative but temporal coefficients (scores) in Table 5 are positive. I cannot fully understand the results of PCA.

18. I suggest that the authors do not interpret the results fully and not always adequately. If I understand correctly, interannual variations of temperature and precipitation are not reflected in the results of PCA. There are presented only mean monthly variability. Consequently, the results of current PCA reflect spatio-temporal variability of monthly mean values. Therefore, the interpretation of the results on page 5 is not valid in the following sentences: "This means that higher values of PC1 correspond to warmer winters …" (lines 15-16), "In general, higher values of PC2 correspond to earlier phenological processes" (lines 28-29), "This means that high PC3 values correspond to overall high precipitation and warm spring, or in other words – overall wetter year" (lines 31-32). If interannual variability is not included into the analysis, the relationships with phenological phases are not appropriate. Anyway, here are many problems to be clarified.

19. It is not clear how the loadings were used to calculate climate indices for the future. I am not sure but it would be correct to realise PCA for the modelled mean values for 2071-2100 and analyse the results of past and future analyses.

20. The section of discussion is pure. I recommend to restructure the paper. In the section of results, there could be only the description of the results of PCA. All interpretations might be included into the section of discussion.

---

## Referee Comment (RC2) · Anonymous Referee #2 · 19 May 2017

In this paper the authors use PCA to derive climate indices for the Baltic states. Rather than use observations directly, the authors incorporate RCM data into the analysis to investigate the effect of climate change on their indices.

The abstract states that the authors are interested in the spatial structure of climate in the study region. Given this aim, the authors chose a suitable method. However, the interpretation of the results is often (in my view) incorrect and the discussion touches repeatedly on the importance of the seasonal cycle. Given this, I suggest the authors modify their aim and conduct a more conventional PCA analysis. I will return to this point below.

As an example of the problems in interpreting the results:

page 3, line 20, interpretation of Figure 4. "more precipitation in, e.g., December is

clearly linked to more precipitation in January. Thus, entire winters are either dry or humid"

This reads as if you are implying some temporal correlation (a dry December is linked to a dry January following), whereas what you really mean is that the spatial precipitation patterns are similar in these months.

"Such relationship between PC1 and original variables implies that high values of PC1 describe climate in which seasons are more similar to each other. "

I do not agree that PC1 represents the magnitude of the seasonal cycle. The seasonal cycle was removed from the 1961-1990 dataset during the normalization. The distinct annual cycle in the PC1 loadings is because the main spatial variation (distance from the coast/continentality) varies seasonally. However, this annual cycle is NOT the same as the seasonal cycle of temperature and precipitation, as per the "normal" definition. For example, the temperature loadings for PC1 (Table 4) have an annual cycle with max in Dec/Jan and a min in May. The seasonal temperature cycle (approximately from Fig 3) has a min in Jan/Feb but a max in July. The difference is clearer for precipitation, where the PC1 precipitation loadings have a max in November and a min in May/June, whereas the precipitation seasonal cycle has a max in August and a min in February. That is, the annual cycle of the precipitation distance-to-the-coast effect is nearly orthogonal to the season cycle of precipitation itself - high values of PC1 do not describe a climate in which precipitation is similar throughout the year! This, for me, is actually the most interesting result from the paper! I can offer no physical explanation for this, and I hope you receive some ideas from this discussion forum! Perhaps you could plot the PC1 loadings on the same graph as the normalized seasonal cycle to demonstrate the result.

Given the above interpretation – that PC1 represents the annual cycle of the distanceto-the-coast effect - I do not believe that the 1961-90 means and variances can be used to normalize the 2071-2100 data. To illustrate, consider a hypothetical climate change
that results only in a change in the temperature seasonal cycle (that is, the change in all temperature grid cells for each month is a constant, but the changes are different for each month). That is, there is no change in the spatial structure in the data. But because the temperature season cycle is not orthogonal to the temperature distance-to-the-coast annual cycle, this change would cause "PC1" 2017-2100 (as calculated in the paper) to differ from PC1, 1961-90. I do not believe interpreting the difference as a change in the continentallity is valid if there is no change in the spatial structure of the dataset!

It is difficult for me to conceptualize the value of a purely spatial climate index for a small region, and the numerous references to the importance of seasonality (mean start date of winter, phenological events, spring floods) suggests that the authors have similar difficulties.

I would suggest that the results would be much easier to interpret if the precipitation data were simply scaled (as one distribution, all months and grid cells) to have the same variance as the temperature data, then the combined dataset normalized (again as one distribution). The first set of loadings will then be the average annual cycle for precipitation and temperature, which is the most important component of any climate index from the point of view of applications. Scaling to the future data in the same way will allow a direct estimate of changes in seasonality.

If the current scaling is retained, the discussion must focus on spatial aspects of the PCs, and not the temporal aspects of the loadings. How do the PCs correlate with distance-to-the-coast? To elevation, slope and aspect?

Finally, the methods section is missing important information:

What interpolation method was used? What grid did you interpolate onto?

what resolution ENSEMBLES simulations were used? did you bias-correct each pixel from each RCM separately? How did you deal with cases where there are more than
one station corresponding to a given RCM pixel (if you used 50km resolution simulations, this must have happened a lot?)

Table 4, 5 and Figure 7 contain essentially the same information; not sure it is worth having all three.

**ESDD**

---

## Author Response (AR2)

Thank you for the reviews. When writing the revised version we carefully considered each of the comments made by referees. This file contains point-by-point response of initial referee reviews and in orange are written additional comments explaining how we addressed the comment in revised version of the paper.

**Referee #1**

**1.** The abstract is too general. I would like to see more concrete results of the study in the abstract.

**Will be considered in revision.**

We added additional details to abstract both clarifying methodology and expanding on results.

**2.** Page 1 line 29. What does mean here the term "homogeneous"? How can the results be more homogeneous? In which sense?

**In the mentioned article (Netzel and Stepinski, 2016) it was obtained that "we demonstrate that clustering-based classification results in climate types that are internally more homogeneous and externally more distinct than climate types in the KGC". Homogenity was measured as "For each class the entropy of a histogram of it constituent types measures a level of class homogeneity with respect to types. Homogeneous classes (like A) have small values of entropy and inhomogeneous classes (like D) have large values of entropy"**

We don't think that introduction of the article should include definition of homogenity. So no changes were done regarding this comment.

**3.** Page 2 lines 3-5. This sentence is a bit unclear and confusing to me. It is written the loadings of components are the coefficients that define indices. I have an imagination that loadings of principal components show correlation between time series of the components and observed variables. Can you explain this?

*"The loadings of chosen principal components are the coefficients that define the newly created indices, which then describe the main features of climate."*

**We have chosen to use most common terminology (at least this is mentioned as most common terminology in Wilks, 2006. Statistical methods in the atmospheric sciences. This book is not mentioned in references, as this terminology is also used in book by Jolliffe, 2002, that we use).**

**Values of principal components are calculated as P = Xe, where X is data matrix of initial data, e are eigenvectors or loadings and P are principal components. Explained variance is calculated from eigenvalues.**

**This means, that correlation coefficients are: corr.coef = loadings * sqrt(eigenvalues).**

**Also we will specify used terminology in revision.**

We have explained methodology in more detail and clarified the terminologu that we use.

**4.** Page 2 line 24. Should it be the reference to de Castro et al. 2007?

*"RCM models are continuously improving and correspond rather well to climate observations (Castro et al. 2007)."*

**We will revise to make it accurate.**

There indeed was mistake in references, this part of the article has been revised and corrected.

**5.** The introduction is lacking of the description of similar studies. PCA is widely used in climatology and also for determining of various climate indices. I suggest a literature overview where is shown how the current study is fitting into other similar studies. The novelty of this study should be clearly indicated.

**Will be considered in revision.**

We feel that we have covered most notable articles in regard to each of the aspects of our study in initial version of the paper already:

- PCA application in general;

- PCA methodology;

- PCA use in index development;

- PCA in Baltic countries;

- use of RCM data.

Our publication is new in combining each of these aspects in a way that hasn't been done before. No changes were made to the paper regarding this comment.

**6.** The description of the use of model data could be more precise. Does the ensemble data mean that the averages of 22 model runs were calculated? What was the spatial resolution of the ensemble data? I suppose that the resolution was different for every single model run.

**We will revise to make it accurate.**

The processing of ensemble data is fully based on publication by Sennikovs and Bethers, 2009 and process is described in more detail there. When revising the paper we decided to emphasize more the reference in our paper instead of fully copying parts of referenced paper.

**7.** It was not clear why the model data were used instead of station data. The density of meteorological stations is rather high in the study area. Therefore, the results of the PCA of station data would be compared with the results of the RCM-based data.

**Main reason was because we have model data for future period. Also model data was bias corrected based on station data so statistics for each case (and also PCA result) should coincide.**

No changes were made in the paper regarding this comment.

**8.** Page 3 line 9. It is not indicated from which source the observation data (Fig.1) were obtained.

**Bias-correction was fully described in (Sennikovs and Bethers 2009). We are considering to remove the illustration of locations and put more emphasis on reference.**

See answer to comment 6.

**9.** Page 3 line 18. Usually, it is written "…as it is done by Malmgren et al. (1999) and Forsythe et al. (2015)".

**Will be considered in revision.**

Was corrected as suggested.

**10.** Page 3 lines 22-23. A very strong correlation in winter precipitation is detected. Is it so that correlations are calculated using the data from the same year? In that case there is not any time lag. I don't believe that there is a correlation above 0.8 between January and December of the same year. I don't believe the statement that winters are either dry or humid. There should be something wrong. I did some calculations with station data of monthly precipitation and did not find any significant correlations. Correlations presented in Fig. 4 are inadequate. Such high correlations in monthly precipitation are not possible at all. All other correlation coefficients seem also suspicious. The reason for presenting the correlation Matrix is not clear.

**We use climatic variables – 30 year average. Data matrix that is used for both correlation calculation and PCA consists of 24 variables (12 temperature and 12 precipitation values) and 7143 cases (grid points). It means that we are looking for spatial patterns, not temporal. This approach is less used, but there are similar applications in literature (for examples, Fovell and Fovell 1993).**

**The reason for correlation matrix was to show that there is redundant information that should be reduced through PCA.**

There was no mistake in our calculations and we have clarified the methodology in the paper.

**11.** Page 4 line 18. I suggest the word "them" instead of "then".

**We will revise to make it accurate.**

Was corrected as suggested.

**12.** Standardisation of climatic data is a trivial procedure. It is not clear why the variances on tables 2 and 3 are presented in the study. Were they spatial mean variances?

**Standardization of data is an important part of PCA that can influence acquired result, and as we are using a bit different approach from the standard (standard meaning subtracting the mean of variable and dividing by square root of variance) it is important to both clarify what we're doing and why. Main reason for tables 2 and 3 was to show the variances for**

each variable and the mean variance per variable category (temperature or precipitation) after we had performed or standardization procedure. Table 2 illustrates what we tried to accomplish (and succeeded) in our standardization procedure. Table 3 illustrates differences in future data (from reference period data in Table 2) in regard to variance of data. Table 3 shows increased precipitation data variance and reduced temperature variance in comparison to reference period. Also Table 3 shows that pattern of variances between different months haven't changed (I was considering illustrating this point, but this would just duplicate information in Tables 2 and 3).

I hope that our explanation has clarified everything. In paper changes were minor, we slightly emphasized the impact of standardization on PCA.

**13.** I think that more information about the PCA procedure is needed. Was it rotated or non-rotated PCA? There are different modes of PCA: T-mode, S-mode etc. How the matrix was performed? Which were variables and which were cases? What were loadings and what were scores? This information is needed for the interpretation of results.

**Will be elaborated in revision. We used non-rotated PCA, and matrix is described in point 10 of this document. Terminology we used is shortly summarized in point 3 of this document. (Principal components = scores. Loadings = eigenvectors). Also we will consider specifying used terminology and technique in revision.**

Additional description of methodology was added to the paper.

**14.** Why the loadings of the three first components are presented in Table 4. What do they show?

**They define approach of using PCA results as climate indices. (This is elaborated a bit more in point 19 of this document).**

We have described methodology and terminology of index calculation in more detail.

**15.** The spatial patterns of three first components are very interesting and informative. I think that they could be wider and better interpreted. It is clear that PC1 represents the influence of the Baltic Sea. It is the main factor causing spatial climatic differences in the Baltic countries. It is directly related to higher temperature and precipitation in autumn and winter, and lower temperature and precipitation in spring and early summer in the coastal regions. In the hinterland far from the sea the spatial coefficients (scores ?) are negative. In conclusion, PC1 reflects continentality of climate.

**Useful input, will be taken into account in revision. It is important to note that values itself (negative, positive, etc.) don't have a meaning associated with them! For example our standardization process of subtracting mean resulted in negative values for precipitation. If we used a different approach, for example, subtracting minimum value, we would get different values of principal components (scores). However that wouldn't change interpretation (correlation coefficients) of PCA results or spatial pattern. What we can compare are regions (or their lack) with similar values.**

Some of the points made by referee regarding the interpretation were included in the revised version of the paper.

**16.** PC2 reflects the second main factor in formation of climate – i.e. latitude. The pattern shows positive scores in Lithuania and negative scores in Estonia. The southern region is characterised by higher temperature, especially in spring and autumn, comparatively higher precipitation from April to June and lower precipitation during the rest of a year.

**Useful input, will be taken into account in rewrite. About positive/negative values see point 15 of this document.**

Some of the points made by referee regarding the interpretation were included in the revised version of the paper.

**17.** The spatial pattern of the PC3 is very similar to the mean annual distribution of precipitation in the study region (Jaagus et al. 2010). Two regions with higher precipitation are described by areas of negative coefficients - one in western Lithuania and Latvia, and another in the western part of continental Estonia and central Vidzeme upland in Latvia. Positive areas correspond to coastal regions with lower precipitation in Estonia and Latvia. But I cannot understand why spatial coefficients (loadings) on Fig. 6 are negative but temporal coefficients (scores) in Table 5 are positive. I cannot fully understand the results of PCA.

**Useful input, will be taken into account in rewrite. Due to standardization some values of precipitation are negative, that can result in negative coefficients. About positive/negative values see point 15 of this document.**

Some of the points made by referee regarding the interpretation were included in the revised version of the paper.

**18.** I suggest that the authors do not interpret the results fully and not always adequately. If I understand correctly, interannual variations of temperature and precipitation are not reflected in the results of PCA. There are presented only mean monthly variability. Consequently, the results of current PCA reflect spatio-temporal variability of monthly mean values. Therefore, the interpretation of the results on page 5 is not valid in the following sentences: "This means that higher values of PC1 correspond to warmer winters …" (lines 15-16), "In general, higher values of PC2 correspond to earlier phenological processes" (lines 28-29), "This means that high PC3 values correspond to overall high precipitation and warm spring, or in other words – overall wetter year" (lines 31-32). If interannual variability is not included into the analysis, the relationships with phenological phases are not appropriate. Anyway, here are many problems to be clarified.

**We will rephrase our interpretations in revision to avoid confusion.**

We have carefully reviewed our interpretation of the results and clarified it wherever possible – mostly emphasizing that correlation coefficients and PCA results describe differences between locations (similarities/differences are spatial not temporal).

**19.** It is not clear how the loadings were used to calculate climate indices for the future. I am not sure but it would be correct to realise PCA for the modelled mean values for 2071-2100 and analyse the results of past and future analyses.

**One of the aims of this work was to see how our components (climate indices) change in future.**

I will try to better explain reasoning for method used in this paper. So the idea is that we perform PCA and acquire principal components, and then once we have some kind of interpretation of principal component, we can just assume that it's a climate index. Try to discard for a moment the principal componets part and just think about the climate index

$$CI = a_1 T_1 + \cdots + a_{12} T_{12} + a_{13} P_1 + \cdots + a_{24} P_{12}$$

We have these coefficients $a_1 \ldots a_{24}$ and we can just calculate this index for present and future.

An analogy would be for example growing degree days. If we want to compare change of growing degree days in present and future we would want to use same base temperature and calendar period to calculate the sum of growing degree days. Even though there might be an argument that in future there is shift in seasons or plant adaptation that would affect methodology. Similarly in our case, we wish to use the same coefficients $a_1 \ldots a_{24}$ for reference period and future, because only then can we make conclusions about the change in value of chosen climate index (It's important to note that currently we can make conslusions about general increase of climate index value or identify regions that have similar values. However we can't make any conclusions about numerical increase, for example, what is the meaning of increase of 5 or 10 units) . Because of the issue about comparison also there were some considerations about standardization process (especially application to future).

We have described methodology and terminology of index calculation in more detail.

**20.** The section of discussion is pure. I recommend to restructure the paper. In the section of results, there could be only the description of the results of PCA. All interpretations might be included into the section of discussion.

**We acknowledge the problem and will consider in it revision.**

We have revised and restucturized 'Results' and 'Discussion' part of the paper.

**Referee #2**

**Responses to the comments:**

    **1. It was mentioned by the referee that scaling should be reconsidered.**

Removal of seasonal cycle doesn't impact our result as PCA is performed on covariance matrix. For example, let's look at two variables X and Y. Let's assume we are performing some standardization on each of them and acquiring new variables X' and Y':

$$X' = \frac{X - C_1}{S_1}, \quad Y' = \frac{Y - C_2}{S_2}$$

Covariance between initial variables is:

$$Cov(X, Y) = \frac{\sum_{i=1}^{n}(X_i - \bar{x})(Y_i - \bar{y})}{n - 1}$$

Covariance between transformed variables:

$$Cov(X', Y') = \frac{\sum_{i=1}^{n}(X_i' - \bar{x}')(Y_i' - \bar{y}')}{n - 1}$$

Where:

$$\bar{x}' = \frac{1}{n}\sum_{i=1}^{n}X_i' = \frac{1}{n}\sum_{i=1}^{n}\left(\frac{X_i - C_1}{S_1}\right) = \frac{1}{S_1}\left(\frac{1}{n}\sum_{i=1}^{n}X_i - C_1\right) = \frac{1}{S_1}(\bar{x} - C_1)$$

And:

$$(X_i' - \bar{x}') = \frac{1}{S_1}(X_i - C_1 - \bar{x} + C_1) = \frac{1}{S_1}(X_i - \bar{x})$$

Similarly:

$$(Y_i' - \bar{y}') = \frac{1}{S_2}(Y_i - \bar{y})$$

And this implies that covariance and therefore PCA is not affected by subtracted values:

$$Cov(X', Y') = \frac{\sum_{i=1}^{n}(X_i - \bar{x})(Y_i - \bar{y})}{(n - 1)S_1 S_2}$$

Subtraction of mean values is important for visualization as it gives similar range for principal components and therefore of their illustration. The change of scaling in regard to subtraction of mean value will impact the values, but won't impact the pattern.

As for scaling future – we wish to compare the change in climate index from past to future and data processing should be kept similar in our case. As a similar example could be considered comparison of growing degree day sum in past and future. If we wish to make

comparison then usually same base temperature and period (start, end date) should be used for both past and future, even though there might be changes in seasonality.

Also our method was mentioned as valid in Cai et al. (2013). Still we agree that adjustments to scaling should be considered (in addition to rotation of principal components) in future development of these climate indices and this point was mentioned in the 'Discussion' part.

We have improved methodology description and description of motivation for used scaling methodology to avoid confusion.

**2. Result interpretation and conclusions**

We agree that the interpretation of our results should be reviewed and often clarified or reconsidered, especially that it should be emphasized that any correlation we mention is spatial not temporal. Many of referee's comments are on point and will be implemented in revised version of the paper.

**"The difference is clearer for precipitation, where the PC1 precipitation loadings have a max in November and a min in May/June, whereas the precipitation seasonal cycle has a max in August and a min in February. That is, the annual cycle of the precipitation distance-to-the-coast effect is nearly orthogonal to the season cycle of precipitation itself – high values of PC1 do not describe a climate in which precipitation is similar throughout the year!"**

[Figure]

The aim of PCA is to explain variance, so there is high correlation with seasonal variance, not seasonal mean values.

We have reviewed description and interpretation of correlation calculation and PCA results and clarified that we are talking about spatial not temporal correlation and differences/similarities. We think that with these changes the aim and results of the paper agree with each other and no significant changes are required for either aim or methodology.

In addition referee mentioned the orthogonality between precipitation seasonal cycle and PC1. However as we have shown in previous response (first point), seasonal cycle doesn't

**impact PC1 loadings, so independence can be expected. Instead loadings are based on the variances (as reflected in the graph above).**

3. **Details on interpolation methodology. "Finally, the methods section is missing important information: What interpolation method was used? What grid did you interpolate onto? what resolution ENSEMBLES simulations were used? did you bias-correct each pixel from each RCM separately? How did you deal with cases where there are more than one station corresponding to a given RCM pixel (if you used 50km resolution simulations, this must have happened a lot?)"**

This is explained in detail in the article that we referenced: Sennikovs, J., Bethers, U.: Statistical downscaling method of regional climate model results for hydrological modelling, Proc.18th World IMACS / MODSIM Congress, Cairns, Australia 13-17, 2009.

We will consider adding more information about the methodology in the revised version of the article.

**The processing of ensemble data is fully based on publication by Sennikovs and Bethers, 2009 and process is described in more detail there. When revising the paper we decided to emphasize more the reference in our paper instead of fully copying parts of referenced paper.**

4. **"Table 4, 5 and Figure 7 contain essentially the same information; not sure it is worth having all three."**

Will be considered in the revised version. There is purpose for each table/figure. Table 4 defines climate indices (this table probably can be considered for removal). In table 5 correlation coefficients are calculated. They are similar to loadings, but there are differences that make them more suited for interpretation of the results. And figure 7 is required to show that acquired climate indices hold there meaning also in future (alternative was table of correlation coefficients for future, but figure was both more illustrative and felt less redundant). We will consider adding more explanation in revision of paper on use of correlation coefficients.

**We have added more explanation of methodology and differences between loadings, correlation coefficients etc. This should clarify usefulness of each table.**

**Major changes:**

1.  More comprehensive description of methodology is added to clarify terminology and avoid confusion.

2.  Method of calculating climate indices has been expanded and clarified.

3.  Restructuration of results and discussion parts of the article has been done.

4.  Clarification of results and their interpretation (main purpose: emphasize that spatial pattern is detected and analyzed) has been done.

5.  We noticed a mistake in Figure 5. (previously Figure 6.) and the figure was replaced. There is no change in any conclusions linked to that figure;

6.  Referees raised several questions in regard to methodology of RCM data use. The method is fully based on the article by Sennikovs and Bethers (2009) and is referenced in the paper. Figure 1 (bias correction locations) is removed as article by Sennikovs and Bethers contains similar, but more informative figure.

[revised manuscript text omitted]

---

## Author Response (AR3)

First, I would like to once more thank the editor and referees about all the input. Second, this file consists of 3 following parts:

- List of main changes;
- Response to referee comments;
- Marked up version of paper with changes.

**List of main changes:**

- Additional clarification and discussion of interpretation regarding PC1;
- Clarification that we use median of RCM ensemble;
- Proofreading.

**Response to referee comments:**

**Referee #1**

I have only two criticisms.

1. It remains unclear to me how the the results from the bias-corrected ensemble were used in the PCA.
Does the PCA use the ensemble median (ie the results shown in Fig 1 & 2)?

2. Interpretation of chagnes in indices from PC1.

>> All indices have higher values in future climate. This can be interpreted as lower difference between seasons (increase of PC1),

Even with the much clearer description of the pca method and terminology, I am still unable to see how this conclusion can be justified. Surely, equation 3 implies that if the temperature data for all months were to increase by one standardized unit, then the future PC1 indices would increase by 1 x sum(T1:T12) (where T1:T12 are values for PC1 from Table 4), ie a change of ~1.1. That is, I do not see how a change in the PC1 indices can be taken to imply a change in seasonality when it appears a non-seasonal (constant) change would also affect the PC1 future indices? Surely you have to subtract the spatial mean of the future data from the future data, rather than use the means from the present-day data, in the standardization? I feel the authors must address this question.

**Response: These both are valid points and we have addressed them in the revised version of the paper.**

1. **Yes, we use median and clarification was added;**
2. **Indeed, this is a good remark that we examined further and added some discussion to the paper.**

**Referee #2**

In my first review I wrote 20 comments or questions. Unfortunately, the authors did not answer exactly but they responded very generally what they have done and changed in the manuscript. Usually it is so that the authors should answer the every single comment. I had several questions that I did not understand. I had some interpretations that I discussed. I did not agree with some interpretations and conclusions made by the authors. Therefore, I would like to continue my discussion with the authors. I ask authors to answer to my questions. What they think and do they agree with my critical remarks and interpretations. It is not a large work to do. The authors have improved the manuscript very much but now I am not able to judge if it is acceptable or not.

**Response: We hope that response in the interactive discussion and later point-by-point response sufficiently addresses all questions.**

[revised manuscript text omitted]